# Improving Patient Outcomes While Reducing Empirical Treatment with Multiplex-Polymerase-Chain-Reaction/Pooled-Antibiotic-Susceptibility-Testing Assay for Complicated and Recurrent Urinary Tract Infections

**DOI:** 10.3390/diagnostics13193060

**Published:** 2023-09-26

**Authors:** Emery Haley, Natalie Luke, Howard Korman, David Baunoch, Dakun Wang, Xinhua Zhao, Mohit Mathur

**Affiliations:** 1Department of Clinical Research, Pathnostics, Irvine, CA 92618, USA; ehaley@pathnostics.com (E.H.); nluke@pathnostics.com (N.L.); 2Department of Urology, Comprehensive Urology—A Division of Michigan Healthcare Professionals, Royal Oak, MI 48073, USA; hkorman@urologist.org; 3Department of Research and Development, Pathnostics, Irvine, CA 92618, USA; dbaunoch@pathnostics.com; 4Department of Scientific Writing, Stat4Ward, Pittsburgh, PA 15238, USA; dakun.wang@stat4ward.com; 5Department of Statistical Analysis, Stat4Ward, Pittsburgh, PA 15238, USA; hannah.zhao@stat4ward.com; 6Department of Medical Affairs, Pathnostics, Irvine, CA 92618, USA

**Keywords:** urinary tract infection (UTI), recurrent UTI (rUTI), complicated UTI (cUTI), multiplex polymerase chain reaction (M-PCR), pooled antibiotic susceptibility testing (P-AST), standard urine culture (SUC), UTI treatment outcomes, UTI-related outcomes

## Abstract

This study compared rates of empirical-therapy use and negative patient outcomes between complicated and recurrent urinary tract infection (r/cUTI) cases diagnosed with a multiplex polymerase chain reaction or pooled antibiotic susceptibility testing (M-PCR/P-AST) vs. standard urine culture (SUC). Subjects were 577 symptomatic adults (*n* = 207 males and *n* = 370 females) presenting to urology/urogynecology clinics between 03/30/2022 and 05/24/2023. Treatment and outcomes were recorded by the clinician and patient surveys. The M-PCR/P-AST (*n* = 252) and SUC (*n* = 146) arms were compared after patient matching for confounding factors. The chi-square and Fisher’s exact tests were used to analyze demographics and clinical outcomes between study arms. Reduced empirical-treatment use (28.7% vs. 66.7%), lower composite negative events (34.5% vs. 46.6%, *p* = 0.018), and fewer individual negative outcomes of UTI-related medical provider visits and UTI-related visits for hospitalization/an urgent care center/an emergency room (*p* < 0.05) were observed in the M-PCR/P-AST arm compared with the SUC arm. A reduction in UTI symptom recurrence in patients ≥ 60 years old was observed in the M-PCR/P-AST arm (*p* < 0.05). Study results indicate that use of the M-PCR/P-AST test reduces empirical antibiotic treatment and negative patient outcomes in r/cUTI cases.

## 1. Introduction

A recent report found that complicated and recurrent urinary tract infections (r/cUTIs) accounted for over 600,000 hospitalizations at an estimated cost of USD 44 billion in the US in 2018 alone [1]. UTIs are also a leading cause of prescribed antibiotic usage in outpatients [2], with most of these antibiotics being prescribed empirically [3]. However, while simple UTIs are typically managed successfully with empirically prescribed antibiotics in an outpatient setting, patients with r/cUTIs have higher treatment failure rates and poorer outcomes, including UTI recurrence (>25% of women with UTIs will have a recurrence within 6 months [4]), urosepsis (comprises approximately 25% of all sepsis cases [5]), and death (12,000 deaths from UTIs annually in the US) [6]. As stakeholders endeavor to counter the threat of microbial antibiotic resistance, the development of diagnostic tests with increased speed and accuracy is critical to improving patient outcomes and antibiotic stewardship and reducing medical resource utilization and costs [6,7,8,9]. 

Standard urine culture (SUC) has been regarded as the gold standard for UTI diagnostic testing for many years [10,11]. Based on several decades-old presumptions that the urinary tract is a sterile environment, SUC methodology is optimized for the growth of gram-negative bacteria, primarily *Escherichia coli* (*E. coli*), the most commonly identified organisms in acute simple UTIs [12,13,14]. The American Urological Association clinical practice guidelines continue to reflect “culture” as the current diagnostic standard [15]. However, recent studies utilizing expanded quantitative urine culture (EQUC), sequencing-dependent methods, such as 16s rRNA sequencing, MALDI-TOF, and other advanced molecular methods have identified several additional microbial species, such as gram-positive organisms, fastidious microbes, and fungi, which can contribute to urinary microbiome dysbiosis in symptomatic subjects [12,16,17,18,19,20,21,22,23,24,25,26,27,28,29].

A significant proportion of presumptive cases of UTIs will end up with negative or other inconclusive results from SUC, leaving a diagnostic gap for healthcare providers as they manage these cases [12,30,31]. Likely due to SUC’s limitations on sensitivity, recent publications have shown poor outcomes for patients with negative SUC results, in whom with EQUC-cultivated organisms that SUC missed [32,33,34]. In those studies, even many SUC-positive cases experience poor outcomes possibly due to limitations on the detection of polymicrobial infections and inability to assess pooled antibiotic susceptibility [35]. Several publications have now demonstrated that in definitive UTI cases, SUC has low sensitivity, contrary to the supposition that newer tests, such as M-PCR, have low specificity for UTI diagnosis [12,25,30,36,37,38]. This supposition of the low specificity of newer tests seems to be largely based on a belief that SUC is generally accurate in identifying negative cases, which is not supported by current evidence and is shown to be incorrect within the cited works above. With the continuing rise of antibiotic resistance, poor outcomes for patients with complicated UTIs, and elevated costs of admitting these patients to hospitals and other high-resource medical care settings, it is important to ask whether the poor outcomes associated with the low sensitivity of SUC can be addressed with more advanced and improved diagnostic tests. 

Recent studies have clearly demonstrated improved sensitivity and specificity of M-PCR compared with SUC for UTI diagnosis [8,9,33,39]; however, studies demonstrating that the use of M-PCR testing results in improved patient outcomes and altered clinician behavior are lacking. This is an important evidence gap to fill since improved detection is not relevant unless it changes patient management and outcomes. There are also significant questions about how the identification by M-PCR of both uropathogens and resistance genes would be used clinically, especially in cases with multiple organisms present in the urine and considering the significant discordance between genotypic resistance and phenotypic susceptibility results [39].

There are newer tests that address both the issue of low sensitivity of SUC and the lack of phenotypic antibiotic susceptibility with M-PCR. In this study, we examine the clinical utility of a novel diagnostic assay that combines M-PCR with pooled antibiotic susceptibility testing (P-AST). This M-PCR/P-AST test provides both genotypic information for pathogen identification and antibiotic resistance and phenotypic antibiotic susceptibility information by testing antibiotics against the pool of viable organisms in the patient’s urine. The test is completed within one day of receipt in the lab and has prior publications demonstrating its association with UTIs [8,9,33,39,40]. 

This prospective, observational study was conducted to compare antibiotic usage and patient outcomes between patients whose treatment was guided by either SUC or M-PCR/P-AST diagnostic test results. Within the two study arms, rates of empirical antibiotic treatment, UTI recurrence rates, and significant negative outcomes were recorded. The aim was to analyze whether the shorter time to results, improved microbial detection, reporting of the presence of antibiotic-resistance genes, and inclusion of pooled phenotypic antibiotic susceptibility results with the M-PCR/P-AST test impacted these important metrics of clinical utility. 

## 2. Materials and Methods

### 2.1. Study Design and Participants

This study was a US-based multi-centered prospective observational study (Clinical trial registration: NCT05091931, https://clinicaltrials.gov/ct2/show/NCT05091931, accessed on 10 July 2023) with Western IRB approval (20214705). The cohort included 577 adult subjects who presented to urology or urogynecology clinics in Southeast Michigan and Kentucky with clinical presentations consistent with r/cUTI between 30 March 2022 and 24 May 2023. The subjects entered either the M-PCR/P-AST arm (*n* = 429) or the SUC arm (*n* = 148), with inclusion and exclusion criteria as described in Appendix A [35].

At the index visit, healthcare providers evaluated subjects’ clinical presentations and ordered urine sample collection and completed a urine test requisition form for SUC or M-PCR/P-AST at their discretion. Each patient’s treatment plan was chosen by the treating clinician in either arm. The treating clinician also answered questions as to the prescribed treatment plan on the medical history form and the treatment decision survey. Study subjects completed a baseline survey on day 1, a daily survey from day 2 through day 14, and a follow-up survey on day 30. Details of the survey questions and study activities are described in Appendix A [35,41,42,43,44].

All the demographic and clinical information was recorded in the REDCap or Castor electronic data capture systems.

### 2.2. Clinical Outcomes and Treatment Status

Clinical outcomes evaluated in this analysis were based on patients’ responses to the clinical outcome questions on the day 30 follow-up survey. Based on patients’ responses to these questions, differences in the following three negative outcomes were measured: (1) the recurrence of UTI symptoms in the index visit; (2) a new visit to a medical provider for UTI symptoms; and (3) a UTI-related hospitalization or visit to urgent care (UC) or the emergency room (ER). The composite negative outcome was defined as patients who had any one or more of these three negative outcomes. 

Treatment status was determined based on the clinical evaluation form completed by healthcare providers and daily surveys completed by the study subjects. “Treated” was defined as treated with antimicrobial agents between day 1 and day 14. Study subjects who did not receive any antimicrobial medication were defined as “Untreated”. For the treated subjects, if the healthcare provider indicated the use of empirical treatment via the responses on the medical history form and the treatment decision survey, the subjects were categorized as “empirically treated”. If the healthcare provider did not indicate the use of empirical treatment, the subjects receiving antimicrobial agents were categorized as “directly treated”.

### 2.3. M-PCR/P-AST Test (Guidance^®^ UTI, Offered by Pathnostics, Irvine, CA, USA)

As described previously [8,9,39], DNA was extracted from the patient’s urine sample using the King Fisher/MagMAX™ automated DNA extraction instrument and the MagMAX™ DNA Multi-Sample Ultra Kit (Thermo Fisher, Carlsbad, CA, USA) per the manufacturer’s instructions. Extracted DNA was then mixed with a universal PCR master mix and amplified using TaqMan technology in a Life Technologies 12K Flex 112-format Open Array System (Thermo Fisher Scientific, Wilmington, NC, USA). Proprietary probes and primers were used to detect 26 bacteria or bacterial groups, fastidious and non-fastidious, and 4 yeast species, as well as 32 antibiotic-resistance genes (see Appendix A for detailed lists).

In addition, fluorescence-based P-AST against 19 antibiotics commonly used for UTIs was performed, as described previously (Appendix A) [35], when at least one non-fastidious bacterium was identified by M-PCR. Briefly, 1 mL of urine specimen was aliquot into a 1.7 mL microcentrifuge tube. The supernatant was aspirated and discarded after centrifugation. The pellet was then suspended with 1 mL of Mueller Hinton Growth (MHG) Media and incubated at 35 °C in a non-CO_2_ incubator for 6 h. MHG Media was used as a negative control. Samples reaching 10,000 cells/mL at the end of the incubation were diluted by aliquoting 0.5 mL of the sample into a 50 mL conical tube containing MHG Media. A 96-well plate pre-loaded with antibiotics was then inoculated with the diluted sample and incubated along with the control plates for 12–16 h at 35 °C in a single layer. Resazurin was added to each well and incubated for 2 h at 35 °C. In the presence of live bacteria, resazurin is reduced to resorufin, which is fluorescent (excitation 530–570 nm; emission 580–590 nm). The fluorescence of each well was measured at 580–590 nm on an Infinite M Nano+ Microplate Reader (TECAN, Männedorf, Switzerland). Growth was indicated by a predetermined fluorescence threshold, as defined by our proprietary assay. Every batch contained reference strain controls as well as blanks to ensure the integrity of the assay.

### 2.4. Standard Urine Culture (SUC)

Standard urine culture was performed as per clinical diagnostic lab standards in laboratories utilized by the participating urology/urogynecology clinic.

### 2.5. Subject Matching

Subject matching for analysis was done based on categories of age, sex, and baseline symptom scores (Figure 1). Out of the 577 total subjects, 398 were successfully matched between the M-PCR/P-AST and SUC arms, with 258 being ≥60 years of age. There were 252 subjects of all ages from the M-PCR/P-AST arm matched to 146 from the SUC arm and 167 subjects ≥60 years of age from the M-PCR/P-AST arm matched to 91 from the SUC arm. Analysis of unmatched subjects is included in the Appendix A.

### 2.6. Statistical Analysis

Patients’ demographics, including age, sex, and baseline FDA symptom scores were summarized using the mean and standard deviation (SD) for continuous variables and frequency (proportion) for dichotomized variables. The chi-square test or Fisher’s exact test was used to test whether the demographics and clinical outcomes differed according to the study arm. All the statistical analyses were performed using Statistical Analysis System (SAS) 9.4. All *p*-values less than 0.05 were considered statistically significant.

## 3. Results

### 3.1. Subject Demographics (Table 1)

Among the 577 total subjects, a total of 398 individuals were successfully matched between the M-PCR/P-AST (*n* = 232) and SUC (*n* = 146) arms based on sex, age, and baseline symptom scores. Demographics of all 577 unmatched subjects can be found in Appendix A. The matched cases were 71% female, and the resulting mean ages and baseline symptom scores were similar between the M-PCR/P-AST and SUC arms. Both total matched cases and cases ≥60 years of age were compared. Matched cases ≥60 years were approximately 69% female, with an average age of 72 years.

**Table 1 diagnostics-13-03060-t001:** Demographics of the Matched Study Cohort.

	All Subjects (*n* = 398)			Subjects ≥60 Years of Age (*n* = 258)		
	SUC (*n* = 146)	M-PCR/P-AST (*n* = 252)	*p*-Value	SUC (*n* = 91)	M-PCR/P-AST (*n* = 167)	*p*-Value
Age			0.39			0.92
Mean (SD)	63.0 (14.8)	64.3 (13.7)		72.3 (7.6)	72.2 (6.9)	
Median (Min, Max)	65.5 (20.6, 96.0)	67.0 (22.2, 95.5)		72.2 (60.3, 96.0)	72.1 (60.5, 95.5)	
Sex (*n* (%))			0.97			0.90
Male	42 (28.8%)	73 (29.0%)		29 (31.9%)	52 (31.1%)	
Female	104 (71.2%)	179 (71.0%)		62 (68.1%)	115 (68.9%)	
Mean Baseline Symptom Score (SD)	5.0 (2.6)	4.8 (2.4)	0.45	5.4 (2.6)	5.0 (2.4)	0.18

### 3.2. Turnaround Time Comparisons between the SUC and the M-PCR/P-AST Arms

When comparing the turnaround time, the time from sample receipt by the lab to the return of results, the M-PCR/P-AST results were returned significantly faster, at approximately half the time of the SUC results (*p* < 0.0001) (Table 2 and Appendix A).

The difference was even greater when considering only samples with positive microbial identifications, where SUC took approximately 3.5 days (Table 3 and Appendix A). 

### 3.3. Clinical Outcome Comparisons between the SUC and the M-PCR/P-AST Arms

We compared a composite of three negative outcomes (recurrence of UTI symptoms; visits to a medical provider for UTIs; and visits to UC, visits to an ER, or hospital stays for UTIs) between subjects whose specimens were tested by SUC and subjects whose specimens were tested by M-PCR/P-AST. All outcomes were based on the patients’ responses to questions on the day 30 follow-up survey.

### 3.4. Outcomes for All Matched Subjects

Across all 398 subjects, the M-PCR/P-AST test arm reported significantly fewer (*p* = 0.018) composite negative outcomes than the SUC test arm (Figure 2). The three separate negative outcomes are broken out in Table 4, and differences in hospital visits, UC, and ER visits are also individually detailed in Table 5. The SUC test arm reported more UTI-related visits to medical providers and visits to UC/ER or hospitalizations compared with the M-PCR/P-AST test arm. However, there were no statistically significant differences in the recurrence of UTI symptoms between the two study arms for the total group. Similar results were found among the total non-matched subjects (*n* = 577) (Appendix A).

### 3.5. Outcomes for Matched Subjects ≥60 Years of Age

For the subset of subjects ≥60 years of age, the M-PCR/P-AST test arm reported significantly fewer (*p* = 0.005) composite negative outcomes than the SUC test arm (Figure 2). The three separate negative outcomes are broken out in Table 6, and differences in hospital visits, UC, and ER visits are also individually detailed in Table 7. The SUC test had significantly higher rates of negative outcomes, with more UTI-related visits to medical providers and visits to UC/ER or hospitalizations. In this subgroup, the M-PCR/P-AST arm also reported significantly fewer recurrences of UTI symptoms than the SUC test arm. Similar results were found among the ≥60 years non-matched subjects (*n* = 416) (Appendix A).

### 3.6. Comparison of Percentages of Patients Who Received Empirical or Directed Antimicrobial Treatments in the SUC and M-PCR/P-AST Arms

Among subjects of all ages, there was no significant difference in percentages of patients treated with antimicrobial agents between the two arms (*p* = 0.55). Of treated subjects in the SUC arm, 66.7% were treated empirically, compared with only 28.7% in the M-PCR/P-AST arm (*p* < 0.0001) (Table 8), which is more than double the rate of empirical-therapy use. Similar results were found among the total non-matched subjects (*n* = 577) (Appendix A).

### 3.7. Comparison of Percentages of Patients 60 Years of Age and Older Who Received Empirical or Directed Antimicrobial Treatments in the SUC and M-PCR/P-AST Arms

Among subjects ≥60 years, percentages of patients treated with antimicrobial agents were similar between the two arms (*p* = 0.63). Of those treated with antimicrobial agents, the empirical-treatment rate in the SUC arm was more than double that of the M-PCR/P-AST arm (69.6% vs. 30.3%, respectively, *p* < 0.0001) (Table 9). Similar results were found among the total non-matched subjects (*n* = 416) (Appendix A).

## 4. Discussion

SUC has been a longstanding test used for acute simple and complicated UTIs for decades and, as such, has carried a presupposition of test efficacy. A close look at the current literature, however, demonstrates that major clinical gaps are created when clinicians rely on the result of this test, which can have a significant negative impact on patients with r/cUTIs [45]. There is consensus that a more sensitive and accurate diagnostic test could favorably impact the management of r/cUTIs, which are major drivers of urosepsis, hospital/ER/urgent-care visits, and poor quality of life (especially in the elderly population) and result in a high cost to healthcare resources [6,46,47,48,49]. In addition to the sensitivity deficiencies associated with the use of standard urine culture for microbial identification and quantification in presumptive UTI cases, the ability to provide actionable antimicrobial sensitivity profiles and the turnaround time for results are problematic, leading to increased use of empirical antimicrobial treatment.

Numerous publications have demonstrated the low sensitivity of SUC, its inability to grow many fastidious organisms known to cause UTIs, the poor outcomes associated with negative test results in patients diagnosed with UTIs, and a significant rate of poor outcomes even when an organism is positively identified [8,9,33,39,40]. 

For example, studies have shown that SUC failed to identify many uropathogens detected by EQUC. Price, T.K., et al. (2016) showed that EQUC detected one or more uropathogens in a UTI-symptomatic cohort (*n* = 69), identifying a total of 110 uropathogens, compared with SUC, which only detected 50% (55 out of 110). Following the clinical treatment selected based on SUC, 41% (24 out of 59) still reported no improvement, half of whom had at least one uropathogen detected by EQUC but not SUC [30]. These data show that compared with EQUC, SUC failed to detect microorganisms that may be contributing to UTI symptoms. 

Even for the detection of classical uropathogens, such as *E. coli*, M-PCR may be more sensitive than SUC. Heytens, S., et al. (2017) performed PCR in addition to SUC to detect *E. coli* in urine samples from women with (*n* = 220) or without (*n* = 86) typical UTI symptoms. In the symptomatic group, 95.9% (211/220) were *E. coli* PCR-positive. In contrast, only 80.9% (178/220) were SUC-positive for any uropathogens, including but not limited to *E. coli*, suggesting that SUC missed the detection of *E. coli* in some women [50].

In comparison, the M-PCR component of the novel M-PCR/P-AST test studied here can identify fastidious organisms, yeast, and polymicrobial infections and provides semi-quantitative cell densities for detected uropathogens. There have been many studies showing the increased sensitivity of PCR compared with SUC but relatively few publications demonstrating specificity for the UTI disease state (instead of simply identifying incidental organisms) or improved clinical utility. 

Several recent publications and manuscripts currently in press have demonstrated the accuracy and clinical validity of the M-PCR/P-AST test used here, with results from inflammatory biomarkers, a UTI diagnosis within a specialty urology setting, clinical outcome and antibiotic-resistance changes, and claims data [8,9,32,33,35,39,40].

This study aimed to measure the clinical utility of the novel M-PCR/P-AST assay for both changing provider behavior, which is central to the usefulness of a diagnostic test and reducing significant negative outcome measures compared with standard urine culture. Our study examined the empirical-treatment rates and the 30-day outcomes of adult patients, and a subgroup of those subjects ≥60 years of age, who visited a urology or urogynecology clinic for suspected r/cUTIs.

We demonstrated a significant reduction (>55% faster for positive results) in turnaround time (time for the lab to return results after receipt of the specimen) to approximately 1.5 days for this test compared with the 3–4 days for SUC. Unlike PCR alone, this novel test also includes P-AST, which also returns results within approximately one day and should provide a more familiar and actionable result for clinicians than M-PCR alone. The P-AST component of the test examines the effectiveness of antibiotics against the pool of organisms in the urine together, as opposed to each in isolation as performed by SUC. P-AST provides a closer representation of polymicrobial infections in the bladder, where microbial species’ cooperation, plasmid transfer, and other mechanisms can result in changes to antibiotic resistance that cannot be accounted for with susceptibility testing from microbial colony isolates.

It is likely that the shorter turnaround time and the availability of P-AST results drove the change in prescriber behavior: a significant reduction of over 50% in empirical-therapy use when the M-PCR/P-AST assay was utilized. This reduction was seen in both the total matched set and in the subgroup ≥60 years of age. Importantly, the percentages of patients treated with antimicrobials were similar between the two arms (67.8% vs. 70.6% for the SUC and M-PCR/P-AST arms, respectively, *p* = 0.55), indicating that the use of the M-PCR/P-AST test is not associated with overuse of antibiotic treatments.

We found that for all adult patients, the SUC test arm experienced a significantly higher occurrence (>35% more) of composite negative outcomes, more medical visits for UTIs, and more visits to UC/ER/hospitals. The observed reduction in negative outcomes was even greater (>58%) in the population aged 60 and over, and there were also lower rates of UTI symptom recurrence with M-PCR/P-AST use in this group. These improvements in outcomes in the M-PCR/P-AST arm are likely due to the previously demonstrated improved accuracy of the M-PCR test coupled with the actionable P-AST assay results, provided with a turnaround time twice as fast or more than SUC.

The poor patient outcomes in the SUC arm match that seen in other publications, as detailed above, and demonstrate the need for an improved diagnostic for r/cUTIs. The results of this study are unique in demonstrating a very significant improvement in patient outcomes when the more advanced M-PCR/P-AST assay is utilized for the management of these cases.

## 5. Conclusions

Our study demonstrated that adult patients, especially older adult patients (≥60 years of age), who visited urology clinics for suspected acute cUTIs experienced significantly reduced use (>50% less) of empirical therapy and exhibited a large reduction in negative outcomes when an M-PCR/P-AST assay was used compared with SUC. These significant improvements in patient outcomes coupled with changes in provider behavior highlight the advantages of utilization of this assay in this patient population. The reduction in empirical therapy use without a corresponding increase in the overall use of antimicrobials, coupled with significant reductions in negative patient outcomes, is the goal of antibiotic stewardship efforts. This evidence, that with a more accurate and timely test, providers prescribed more effective antibiotics and were willing to wait for test results before making treatment decisions, is critical to improving the management of recurrent and complicated UTIs and patient health.

## Figures and Tables

**Figure 1 diagnostics-13-03060-f001:**
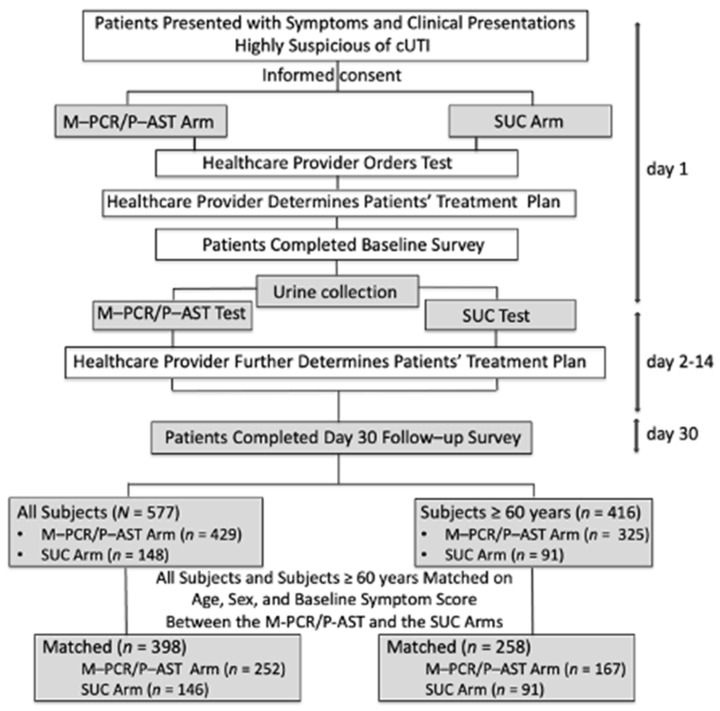
Study flowchart.

**Figure 2 diagnostics-13-03060-f002:**
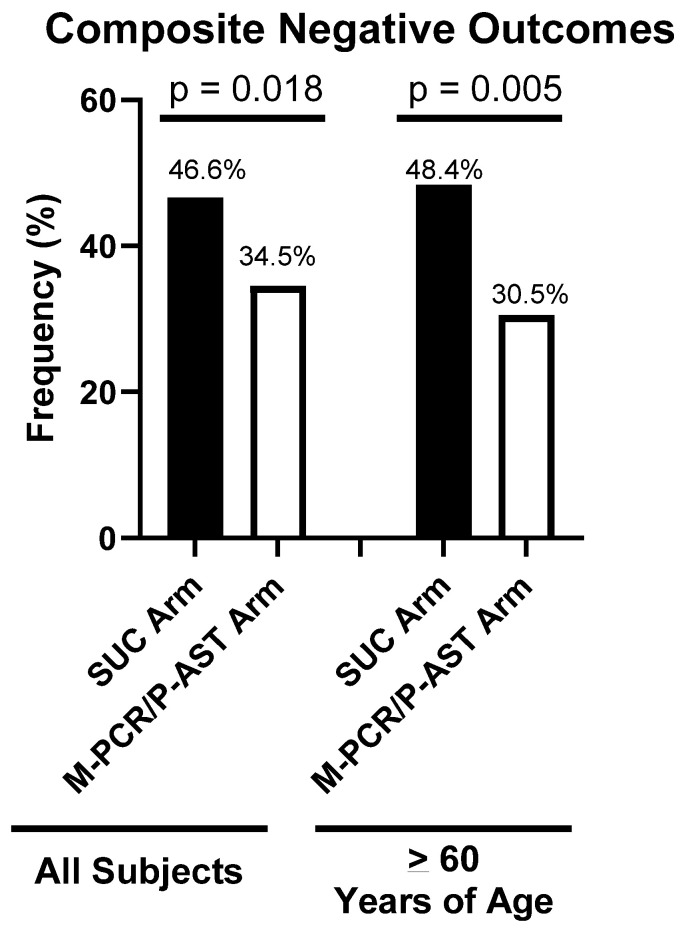
Frequency of Negative Outcomes. Comparison of the frequency of composite negative outcomes for the SUC arm versus the M-PCR/P-AST arm among all subjects and among subjects 60 years of age and older. Bars demonstrate the frequency of negative outcomes as a percentage of the total subjects in each study arm.

**Table 2 diagnostics-13-03060-t002:** Comparison of Turnaround Times between M-PCR/P-AST and SUC for All Matched Subjects.

	SUC	M-PCR/P-AST	*p*-Value
	Mean Days (SD)		
Full sample matched (*n* = 248 M-PCR/P-AST; *n* = 131 SUC)	2.87 (1.14)	1.45 (0.55)	<0.0001
Sample of age 60+ matched (*n* = 164 M-PCR/P-AST; *n* = 81 SUC)	2.94 (1.13)	1.48 (0.56)	<0.0001

**Table 3 diagnostics-13-03060-t003:** Comparison of Turnaround Times between M-PCR/P-AST and SUC for All Matched Subjects with Positive Results.

	SUC	M-PCR/P-AST	*p*-Value
	Mean Days (SD)		
Full sample matched (*n* = 197 M-PCR/P-AST; *n* = 53 SUC)	3.56 (1.14)	1.52 (0.55)	<0.0001
Sample of age 60+ matched (*n* = 134 M-PCR/P-AST; *n* = 42 SUC)	3.54 (1.08)	1.55 (0.56)	<0.0001

**Table 4 diagnostics-13-03060-t004:** Comparison of Negative Outcomes for All Matched Subjects.

	Overall		
	SUC	M-PCR/P-AST	*p*-Value (SUC vs. M-PCR/P-AST)
	*n*/Total (%)	*n*/Total (%)	
Recurrence of UTI symptoms	31/146 (21.2%)	42/252 (16.7%)	0.26
Medical provider visit (for UTI)	49/146 (33.6%)	55/252 (21.8%)	0.010
Hospital, ER, or UC visit (for UTI)	26/146 (17.8%)	27/252 (10.75%)	0.045

**Table 5 diagnostics-13-03060-t005:** Comparison of Individual Negative Outcomes for All Matched Subjects.

Total *n* = 398	SUC *n* = 146 *n* (%)	M-PCR/P-AST *n* = 252 *n* (%)	*p*-Value
Any UTI-related hospital visits	13 (8.9%)	11 (4.4%)	0.07
Any UTI-related ER visits	19 (13.0%)	18 (7.1%)	0.052
Any UTI-related UC visits	19 (13.0%)	14 (5.6%)	0.009
UTI-related hospital visits (number of occurrences)			0.08
0	133 (91.1%)	241 (95.6%)	
1	7 (4.8%)	7 (2.8%)	
2	3 (2.1%)	0 (0%)	
3+	3 (2.1%)	4 (1.6%)	
UTI-related ER visits (number of occurrences)			0.16
0	127 (87.0%)	234 (92.9%)	
1	10 (6.9%)	11 (4.4%)	
2	3 (2.1%)	1 (0.4%)	
3+	6 (4.1%)	6 (2.4%)	
UTI-related UC visits (number of occurrences)			0.03
0	127 (87.0%)	238 (94.4%)	
1	9 (6.2%)	4 (1.6%)	
2	2 (1.4%)	4 (1.6%)	
3+	8 (5.5%)	6 (2.4%)	

**Table 6 diagnostics-13-03060-t006:** Comparison of Negative Outcomes for Matched Subjects ≥60 Years of Age.

	≥60 Years		
	SUC	M-PCR/P-AST	*p*-Value (SUC vs. M-PCR/P-AST)
	*n*/Total (%)	*n*/Total (%)	
Recurrence of UTI symptoms	22/91 (24.2%)	23/167 (13.8%)	0.035
Medical provider visit (for UTI)	29/91 (31.9%)	34/167 (20.4%)	0.040
Hospital, ER, or UC visit (for UTI)	20/91 (22.0%)	14/167 (8.4%)	0.002

**Table 7 diagnostics-13-03060-t007:** Comparison of Individual Negative Outcomes for Matched Subjects ≥60 Years of Age.

Total *n* = 258	SUC *n* = 91 *n* (%)	M-PCR/P-AST *n* = 167 *n* (%)	*p*-Value
Any UTI-related hospital visits	9 (9.9%)	6 (3.6%)	0.039
Any UTI-related ER visits	14 (15.4%)	12 (7.2%)	0.037
Any UTI-related UC visits	14 (15.4%)	6 (3.6%)	0.0007
UTI-related hospital visits (number of occurrences)			0.041
0	82 (90.1%)	161 (96.4%)	
1	5 (5.5%)	2 (1.2%)	
2	2 (2.2%)	0 (0%)	
3+	2 (2.2%)	4 (2.4%)	
UTI-related ER visits (number of occurrences)			0.084
0	77 (84.6%)	155 (92.8%)	
1	8 (8.8%)	7 (4.2%)	
2	2 (2.2%)	0 (0%)	
3+	4 (4.4%)	5 (3.0%)	
UTI-related UC visits (number of occurrences)			0.001
0	77 (84.6%)	161 (96.4%)	
1	7 (7.7%)	1 (0.6%)	
2	2 (2.2%)	3 (1.7%)	
3+	5 (5.5%)	2 (1.2%)	

**Table 8 diagnostics-13-03060-t008:** Comparisons of percentages of patients empirically or directedly treated with antimicrobial agents among the matched subjects in the SUC and M-PCR/P-AST arms.

Total (*n* = 398)	SUC Arm (*n* = 146)	M-PCR/P-AST Arm (*n* = 252)	*p*-Value
Not treated with antimicrobial agents (*n*, %)	47, 32.2%	74, 29.4%	0.55
Treated with antimicrobial agents (*n*, %)	99, 67.8%	178, 70.6%	
Of those treated with antimicrobial agents (*n* = 277)	SUC Arm (*n* = 99)	M-PCR/P-AST Arm (*n* = 178)	
Empirical treatment (*n*, %)	66, 66.7%	51, 28.7%	<0.0001
Directed treatment (*n*, %)	33, 33.3%	127, 71.4%	

**Table 9 diagnostics-13-03060-t009:** Comparisons of percentages of patients empirically or directedly treated with antimicrobial agents among the matched subjects ≥60 years in the SUC and M-PCR/P-AST arms.

Total (*n* = 258)	SUC Arm (*n* = 91)	M-PCR/P-AST Arm (*n* = 167)	*p*-Value
Not treated with antimicrobial agents (*n*, %)	22, 24.2%	45, 27.0%	0.63
Treated with antimicrobial agents (*n*, %)	69, 75.8%	122, 73.1%	
Of those treated with antimicrobial agents (*n* = 191)	SUC Arm (*n* = 69)	M-PCR/P-AST Arm (*n* = 122)	
Empirical treatment (*n*, %)	48, 69.6%	37, 30.3%	<0.0001
Directed treatment (*n*, %)	21, 30.4%	85, 69.7%	

## Data Availability

The data presented in this study are available on request from the corresponding author. The data are not publicly available due to privacy concerns.

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
