# Peer review of "Improving Patient Outcomes While Reducing Empirical Treatment with Multiplex-Polymerase-Chain-Reaction/Pooled-Antibiotic-Susceptibility-Testing Assay for Complicated and Recurrent Urinary Tract Infections"

_diagnostics, 2023, doi:10.3390/diagnostics13193060_

Round 1
Reviewer 1 Report
The authors of the article titled "Improving Patient Outcomes While Reducing Empirical Treatment with M-PCR/P-AST Assay for Complicated and Recurrent UTIs" conducted a study to compare antibiotic usage and patient outcomes between subjects whose treatment was guided by SUC or M-PCR/P-AST diagnostic tests.
Between 03/30/2022 and 05/24/2023, samples were collected from 577 UTI patients and tested using M-PCR/P-AST or SUC tests. The work is generally fine and could add some fresh data regarding human health and antibiotic resistance. The manuscript has good language.
Here are a few comments that, in my opinion, need to be addressed:
1- Methodology, 148 and 429 subjects, respectively, underwent M-PCR/P-AST and SUC testing. How would a significant difference between M-PCR/P-AST and SUC of roughly three times be justifiable given that this is a comparative study?
2- Methodology, What factors were taken into consideration when choosing which test to run on a patient? How were 429 patients shortlisted for the M-PCR test and 148 were for the SUC test?
3- Methodology, Line 167- Please provide information about the primers that were used in the study.
4- Methodology, P-AST- In the P-AST, what fluorescence dye or marker was used? What standards were used to determine the tested bacteria's susceptibility to antibiotics?
5- Results, Table 1– Do you think your results are statistically significant based on the p-value? Please also explain why the p-values in the other cases were insignificant.
6- The discussion section falls short and is unable to adequately convey the significance of the new testing method. Please revise the discussion to compare it to the literature that is currently available. based on previously published data, why the new test method is superior to the old one.
English language is fine.
Author Response
Reviewer 1
The authors of the article titled "Improving Patient Outcomes While Reducing Empirical Treatment with M-PCR/P-AST Assay for Complicated and Recurrent UTIs" conducted a study to compare antibiotic usage and patient outcomes between subjects whose treatment was guided by SUC or M-PCR/P-AST diagnostic tests.
Between 03/30/2022 and 05/24/2023, samples were collected from 577 UTI patients and tested using M-PCR/P-AST or SUC tests. The work is generally fine and could add some fresh data regarding human health and antibiotic resistance. The manuscript has good language.
Here are a few comments that, in my opinion, need to be addressed:
1- Methodology, 148 and 429 subjects, respectively, underwent M-PCR/P-AST and SUC testing. How would a significant difference between M-PCR/P-AST and SUC of roughly three times be justifiable given that this is a comparative study?
Response: Thank you for your question. In our comparative study, which is not a randomized study, participating physicians had the freedom to choose which arm to assign the patients to. To account for any confounding factors between the M-PCR/P-AST arm and standard urine culture arm, patient matching was done, and the analysis was performed on the matched sets. This matching was done based on patient age, sex, and baseline symptom scores, resulting in 252 subjects from the M-PCR/P-AST arm being compared to 146 subjects from the SUC arm, a ratio of approximately 1.7:1. Both the comparison of the matched set and the full cohort (Supplemental data) showed a similar reduction in the negative outcomes in the M-PCR/P-AST arm.
2- Methodology, What factors were taken into consideration when choosing which test to run on a patient? How were 429 patients shortlisted for the M-PCR test and 148 were for the SUC test?
Response: Thank you for your question. As explained in our response to question 1, this is not a randomized clinical study. Since this is an observational study, the participating physicians had the freedom to choose which arm to assign the patients to. Therefore, this sample size distribution between the two arms (429 for M-PCR and 148 for SUC) indicated that physicians participating in the study were more likely to choose M-PCR/P-AST than SUC in the clinical diagnosis and management of UTI. To eliminate selection bias, patients were matched between the two arms based on age, sex, and baseline symptom scores. Further studies that are randomized clinical trials will allow for the assignment of testing to each arm. Clarification that this study was not randomized and that clinicians were able to choose their preferred test has been added to the Methods section and Supplemental Table S1.
3- Methodology, Line 167- Please provide information about the primers that were used in the study.
Response: Thank you for your suggestion. The primers and targets used for each assay are proprietary to ThermoFisher, but they have been validated by ThermoFisher and Pathnostics. We have provided the primer and probe information in the supplemental information (Supplemental Tables S4 and S5).
4- Methodology, P-AST- In the P-AST, what fluorescence dye or marker was used? What standards were used to determine the tested bacteria's susceptibility to antibiotics?
Response: Thank you for the question. Clarification has been added in lines 187-193.
5- Results, Table 1– Do you think your results are statistically significant based on the p-value? Please also explain why the p-values in the other cases were insignificant.
Response: Thank you for your questions. The results were statistically significant for the reduction of negative outcomes, turnaround times, and empiric therapy use as demonstrated by the p-values in almost all comparisons for both those over 60 years and the full cohort. The one exception in the reduction of negative outcomes was the comparison of the recurrence of UTI symptoms in the full cohort, which may be due to the inclusion of younger patients who are generally expected to have better outcomes with UTIs than the elderly. In the population over 60 years, the reduction in symptom recurrence was significant when Guidance® UTI (M-PCR/P-AST) was used. Overall, for composite reduction of negative outcomes and for individual negative outcomes, there was a significant reduction when comparing the two arms.
For Table 1 detailing the comparison of the matched set, there was no statistically significant difference, which indicates that there were no statistically significant differences in the potentially confounding factors of age, sex, and mean baseline symptom score between the patients in the M-PCR/P-AST and the SUC arms. This demonstrated that the two arms had comparable baseline demographics after matching, which is an essential feature for reducing bias in assessing outcomes.
There was also no significant difference seen in the overall use of antibiotics between the two arms, with a p-value over 0.05.
Based on your review comment, we have now added this sentence below in the “statistical analysis” subsection in the “Materials and Methods” section of the manuscript, in lines 210 and 211: “All p-values less than 0.05 were considered statistically significant.”
6- The discussion section falls short and is unable to adequately convey the significance of the new testing method. Please revise the discussion to compare it to the literature that is currently available. based on previously published data, why the new test method is superior to the old one.
Response: We appreciate your feedback. We have detailed some of the cited publications' findings in lines 325-338 in the Discussion, which detail similar shortcomings of SUC vs. more advanced UTI tests. We have also added the language: “The poor patient outcomes in the SUC arm match that seen in other publications, as detailed above, and demonstrate the need for an improved diagnostic for r/cUTIs. The results of this study are unique in demonstrating a very significant improvement in patient outcomes when the more advanced M-PCR/P-AST assay is utilized for the management of these cases” to tie the findings in the paper to others in the recently published literature.
The specific findings in this publication are somewhat unique, since prior work on the clinical utility of advanced UTI tests, such as improved patient outcomes and reduced empiric therapy, is lacking. No prior publications directly measured these outcomes; however, the cited works demonstrate the poor performance of SUC, consistent with that demonstrated in this study.
Reviewer 2 Report
The current study "Improving Patient Outcomes While Reducing Empirical Treat-2 ment with M-PCR/P-AST Assay for Complicated and Recurrent 3 UTIs" elaborated by Emery Haley et al. aims at a very current topic, that of empiric antibiotic therapy and how we can improve this aspect in the more precise treatment of urinary infections and decrease the overuse of some antibiotics that could lead to the increase of antimicrobial resistance. The study is a very well elaborated one, proof of sustained work, with a very well made and described statistic that brings a definite benefit to current knowledge. However, some aspects should be mentioned:
1. I believe that the terms recurrent and complicated should be included in the keywords.
2. Line 40: "(12,000 deaths from UTI annually in the US). [6]" - the reference must be placed before the period. Also, in all other situations.
3. Line 87: "Here, we conducted a prospective,..." - it is preferable in formal, academic language to avoid using the first person; e.g. "The current study presents...", "The study was conducted..."
4. The Materials and method chapter is exhaustive, very complete, but I think it should be narrowed down and reformulated in a more concise framework.
5. In the Results chapter, please do not repeat in several places the information from the tables to be repeated in the text.
6. In the Discussions chapter, comparative data are needed between the results obtained in this paper and other studies carried out in this direction.
Overall, the study is a very good one, with clear results, comprehensive tables and explanatory figures, with the formulation of clear conclusions that can guide clinicians towards a new way of evaluating urinary pathogens and their susceptibility to antibiotics in order to reduce the empiric administration period of antibiotics. I believe that the article can be published after minor revision.
Author Response
Reviewer 2
The current study "Improving Patient Outcomes While Reducing Empirical Treatment with M-PCR/P-AST Assay for Complicated and Recurrent 3 UTIs" elaborated by Emery Haley et al. aims at a very current topic, that of empiric antibiotic therapy and how we can improve this aspect in the more precise treatment of urinary infections and decrease the overuse of some antibiotics that could lead to the increase of antimicrobial resistance. The study is a very well-elaborated one, proof of sustained work, with a very well-made and described statistic that brings a definite benefit to current knowledge. However, some aspects should be mentioned:
- I believe that the terms recurrent and complicated should be included in the keywords.
Response: Thank you for your suggestion. We have added recurrent UTI and complicated UTI in the keywords.
- Line 40: "(12,000 deaths from UTI annually in the US). [6]" - the reference must be placed before the period. Also, in all other situations.
Response: Thank you for your suggestion. We have corrected the placement of reference citations in the whole manuscript document.
- Line 87: "Here, we conducted a prospective,..." - it is preferable in formal, academic language to avoid using the first person; e.g. "The current study presents...", "The study was conducted..."
Response: Thank you for your comment. We have made the changes in Line 88 (original Line 87) as you suggested.
- The Materials and method chapter is exhaustive, very complete, but I think it should be narrowed down and reformulated in a more concise framework.
Response: We appreciate your comment and suggestion. The Materials and Methods section (lines 96-213) has been revised to be more concise and added a new supplemental table S1 with the more complete detail.
- In the Results chapter, please do not repeat in several places the information from the tables to be repeated in the text.
Response: Thank you for your helpful suggestion. We have revised the results section to reduce the repetition of information in the tables.
- In the Discussions chapter, comparative data are needed between the results obtained in this paper and other studies carried out in this direction.
Response: We appreciate your constructive suggestion.
We have detailed some of the cited publications' findings in lines 325-338 in the Discussion, which detail similar shortcomings of SUC vs. more advanced UTI tests. We have also added the language: “The poor patient outcomes in the SUC arm match that seen in other publications, as detailed above, and demonstrate the need for an improved diagnostic for r/cUTIs. The results of this study are unique in demonstrating a very significant improvement in patient outcomes when the more advanced M-PCR/P-AST assay is utilized for the management of these cases” to tie the findings in the paper to others in the recently published literature.
The specific findings in this publication are somewhat unique, since prior work on the clinical utility of advanced UTI tests, such as improved patient outcomes and reduced empiric therapy, is lacking. No prior publications directly measured these outcomes; however, the cited works demonstrate the poor performance of SUC, consistent with that demonstrated in this study.
- Overall, the study is a very good one, with clear results, comprehensive tables and explanatory figures, with the formulation of clear conclusions that can guide clinicians towards a new way of evaluating urinary pathogens and their susceptibility to antibiotics in order to reduce the empiric administration period of antibiotics. I believe that the article can be published after minor revision.
Response: We appreciate your positive review.
Round 2
Reviewer 1 Report
The majority of the comments have been addressed satisfactorily by the authors.
Language of the manuscript is satisfactory.